Genome-wide identification and expression analysis of SBP-box gene family reveal their involvement in hormone response and abiotic stresses in Chrysanthemum nankingense

Li Ziwei 1
Yang Yujia 1
Chen Bin 1
Xia Bin 1
Li Hongyao 1
Zhou Yunwei dlzhyw@126.com 2
He Miao hemiao_xu@126.com 1
1 College of Landscape Architecture, Northeast Forestry University , Harbin , Heilongjiang , China
2 College of Horticulture, Jilin Agricultural University , Jilin , China
Azeem Farrukh
Electronic publication date: 2022 Oct 27
Publication date: 2022
Volume: 10
Electronic Location ID: e14241
Received 2022 May 23; Accepted 2022 Sep 23
Copyright: ©2022 Li et al.
Copyright year: 2022
Copyright holder: Li et al.
License: This is an open access article distributed under the terms of the Creative Commons Attribution License, which permits unrestricted use, distribution, reproduction and adaptation in any medium and for any purpose provided that it is properly attributed. For attribution, the original author(s), title, publication source (PeerJ) and either DOI or URL of the article must be cited.
License URL: https://creativecommons.org/licenses/by/4.0/

Keywords: Chrysanthemum nankingense, SBP-box gene, Evolutionary analysis, Expression profile, Hormone response, Abiotic stresses

Funding: The National Key Research and Development Program of China 2018YFD1000406 The National Key Research and Development Program of China 2019YFD1001504 The Natural Science Foundation of Heilongjiang Province, China LH2021C018 This research was supported by the National Key Research and Development Program of China (2018YFD1000406), the National Key Research and Development Program of China (2019YFD1001504) and the Natural Science Foundation of Heilongjiang Province, China (No. LH2021C018). The funders had no role in study design, data collection and analysis, decision to publish, or preparation of the manuscript.

==============================
SQUAMOSA promoter-binding-protein (SBP)-box family proteins are a class of plant-specific transcription factors, and widely regulate the development of floral and leaf morphology in plant growth and involve in environment and hormone signal response. In this study, we isolated and identified 21 non-redundant SBP-box genes in Chrysanthemum nankingense with bioinformatics analysis. Sequence alignments of 21 CnSBP proteins discovered a highly conserved SBP domain including two zinc finger-like structures and a nuclear localization signal region. According to the amino acid sequence alignments, 67 SBP-box genes from Arabidopsis thaliana, rice, Artemisia annua and C. nankingense were clustered into eight groups, and the motif and gene structure analysis also sustained this classification. The gene evolution analysis indicated the CnSBP genes experienced a duplication event about 10 million years ago (Mya), and the CnSBP and AtSPL genes occurred a divergence at 24 Mya. Transcriptome data provided valuable information for tissue-specific expression profiles of the CnSBPs, which highly expressed in floral tissues and differentially expressed in leaf, root and stem organs. Quantitative Real-time Polymerase Chain Reaction data showed expression patterns of the CnSBPs under exogenous hormone and abiotic stress treatments, separately abscisic acid, salicylic acid, gibberellin A3, methyl jasmonate and ethylene spraying as well as salt and drought stresses, indicating that the candidate CnSBP genes showed differentiated spatiotemporal expression patterns in response to hormone and abiotic stresses. Our study provides a systematic genome-wide analysis of the SBP-box gene family in C. nankingense. In general, it provides a fundamental theoretical basis that SBP-box genes may regulate the resistance of stress physiology in chrysanthemum via exogenous hormone pathways.

Introduction

Plants may confront a variety of environmental stresses that adversely affect their growth and productivity, such as extreme temperatures, water-deficiencies, drought and salinity stress (Saibo, Lourenco & Oliveira, 2009). Plants have evolved many mechanisms to overcome abiotic stresses, including the modification of expression patterns in stress-responsive genes for adaptive development and growth (Skirycz & Inze, 2010). Transcription factors (TFs), are groups of important regulatory factors in plants which generally play critical roles in plant growth, differentiation, metabolism mechanism, response to hormone signals and adversity conditions (Liu et al., 2021b; Song et al., 2022). Plant hormones are the center regulators of physiological reactions and biochemical processes in cells, because they not only initiate internal development perception, but also transmit exoteric environmental inputs. (Glazebrook, 2005). The phytohormones, such as abscisic acid (ABA), jasmonic acid (JA), gibberellin (GA), ethylene (ETH), and salicylic acid (SA), integrate environmental stress signaling to mediate the growth and development of plants (Hou et al., 2013; Colebrook et al., 2014).

SQUAMOSA promoter-binding protein (SBP)-box genes encode plant-specific TFs that possess approximately 76 amino acids and a highly conserved DNA-binding domain consisting of approximately 76 amino acid including two typical zinc-finger structures, C3H and C2HC, and a nuclear localization signal region, NLS (Yamasaki et al., 2004; Birkenbihl et al., 2005; Guo et al., 2008). SBP-box genes, AmSBP1 and AmSBP2, were initially discovered in snapdragon (Antirrhinum majus) due to their interactions with the promoter sequence region of the floral meristem identity gene SQUAMOSA (a kind of MADS-box), which are relevant to the origin and evolution of reproductive structures such as flowers and ovules (Klein, Saedler & Huijser, 1996). In higher plants, the transformation from vegetative stage to reproductive stage of life is an important phase during time of flowering. So it is of great significance to explore the functions of SBP-box gene family in chrysanthemum. Since then, SBP-box genes have been isolated and characterized in many plants ranging from the single-celled alga (Chlamydomonas reinhardtii) (Kropat et al., 2005) to model plant, Arabidopsis thaliana (Cardon et al., 1999) and from world-wide cultivated crops like rice (Oryza sativa) (Xie, Wu & Xiong, 2006), Chinese cabbage (Brassica rapa) (Cheng et al., 2016) and wheat (Triticum aestivum) (Li et al., 2020) to fruits like sweet orange (Citrus sinensis) (Song et al., 2021), apple (Malus ×domestica Borkh.) (Li et al., 2013) and sugarcane (Saccharum spontaneum) (Feng et al., 2021).

SBP-box genes regulate many processes of development and floral regulation in flowering plants, including the vegetative phase change (Xu et al., 2016), flowering (Xu et al., 2016), leaf initiation (Preston et al., 2016), shoot and inflorescence branching (Shao et al., 2019; Cui et al., 2020), fruit development and ripening (Ferreira e (Silva et al., 2014)), floral organ development and fertility (Liu et al., 2017b) and pollen sac development (Unte et al., 2003). It previously reported that AtSPL3/4/5 redundantly promoted the floral meristem transition and exhibited early-flowering phenotype by binding to the promoters of LEAFY (LFY), FRUITFUL (FUL), and APETALA1 (AP1), and acted synergistically with the FLOWERING LOCUS T (FT)-FD module to induce flowering under long-day (LD) condition (Yamaguchi et al., 2009). In rice, OsSPL16 participated in the regulation of size, shape and quality of grains (Wang et al., 2012; Yang et al., 2019) and OsmiR156k-OsSPL18-DEP1 module regulated the weight and number of grains (Yuan et al., 2019).

Previous studies have reported that SBP-box genes have a pivotal role in various stresses and hormone signaling pathways (Wang et al., 2009). AtSPL7 and AtSPL14 separately were pivotal participators in response to copper homeostasis and cell death-inducing fungal toxin fumonisin B1 (FB1) (Stone et al., 2005; Yamasaki et al., 2009). Over-expression of AtSPL1 and AtSPL12 enhanced thermos-tolerance during reproductive growth in inflorescence (Chao et al., 2017), and OsSPL10 negatively regulated salt tolerance in rice (Lan et al., 2019). Besides, the VpSBP genes in grape overexpressed in Arabidopsis improved the tolerance of salt and drought coordinate stress in regulation of salt hypersensitivity (SOS) and reactive oxygen species (ROS) signaling cascades (Hou et al., 2018), and CiSPL genes in pecan (Carya illinoinensis) showed apparent spatiotemporal expression patterns under salt and drought treatments (Wang et al., 2021). VvSBP and MdSBP genes in grape and apple may be dependent on hormonal signaling pathway to reveal involvement in regulation mechanism against abiotic stresses (Hou et al., 2013; Li et al., 2013) furthermore PgSPL5 and PgSPL13 were proved to involve in plant hormone signal transduction in development in pomegranate (P. granatum) (Li et al., 2021). CmmiR156-targeted CmSBP genes reduced expression levels via GA signaling pathway in Castanea mollissima (Chen et al., 2019).

MicroRNAs, miRNAs, a class of endogenous non-coding RNAs, 20–24 nucleotides, were proved to target some SBP-box genes and form RNA-induced silencing complexes to regulate functions in plants. In Arabidopsis and rice, 11 of 17 and 11 of 19 SBP-box genes possessed the miR156-targeted sites, which were located in either coding region (CDS) or 3′untranslated region (3′UTR) (Xie, Wu & Xiong, 2006; Xing et al., 2010). Recently, the involvements of miR156-SBP/SPL regulation modules in lots of plant developmental processes and stresses have come to light. MiR156/529/535-SPL gene modules regulated the cereal panicle development and higher cytokinin accumulation in female inflorescence in oil palm (Tregear et al., 2022). MiR156 overexpression inhibited non-targeting SBP mutation through the raise of DELLA and GA-decomposing enzymes, resulting in stronger phenotypes. And GA also coordinated to other hormones to regulate phase transition via miR156-SBP/SPL modules (Jerome Jeyakumar et al., 2020). AthmiR156-targeted SPL13 downregulated to enhance the tolerance of drought (Beveridge & Kyozuka, 2010). Besides, miR156-targeted SPL2/9/11 genes neutralized negative effects of up-regulated miR156 under heat stress in plant growth and TcSPLs in tamarisk showed a critical post-transcription regulation at 1 h under salt stress (Stief et al., 2014; Wang et al., 2019).

Chrysanthemum is famous for its ornamental and medicinal value regarded as one of the most valuable floricultural crops in the world (Zhang et al., 2014). Hybridization and artificial selection extensively exist in genus Chrysanthemum, causing that polyploid species and species complexes create highly diversify in ploidy levels, morphology of flowers and leaves, colors of ray florets and environmental tolerances, which bring about great market demand prospects and valuable genetic resources to chrysanthemum breeding (Ma et al., 2020; Qi et al., 2021). Diversification of growth conditions relatively restrict the development of the native chrysanthemum resources. Also, soil salinity and moisture increase the production capacity consumption in facility cultivation and become the limiting factor of costs. Therefore, it is significant to investigate the resistance mechanism of chrysanthemum. C. nankingense (2n = 2x = 18), a diploid native species of China, processes a key progenitor genomic model (Yang et al., 2006; Ren et al., 2014). The success of the whole C. nankingense genome sequencing is doubtlessly a milestone in the direction of herbaceous plants molecular research, and makes it possible to excavate gene families from genome-wide to provide molecular basis in genetic evolution mechanism (Song et al., 2018). It is well known that SBP-box gene family acts as a pivotal regulatory in formation of some phenotypes and integration of growth and environmental signals. In tea plant, CsSBP genes were response to hormone signals and abiotic stresses, and showed obvious co-expression of CsSBP2/10 across MeJA, SA and salt treatments (Teng et al., 2021). Previous studies mostly focused on flowering mechanism and fruit development, but little known about the potential physiological functions of SBP-box family genes. In this study, we performed genome-wide identification of the SBP-box gene family in C. nankingense, and the characterization, phylogeny, gene structures, miR156-targeted genes and tissue-specific expression analysis were investigated by bioinformatics and experiments. We also endeavored to analyze the expression levels of 21 CnSBP genes under exogenous hormones and abiotic stresses treatments. This research provided a fundamental theoretical basis of candidate hormone- and stress-responsiveness CnSBP genes and further elucidated the potential functions in response to biotic and abiotic stresses dependent on hormone signal pathway.

Materials and Methods

Plant materials and treatments

The seeds of C. nankingense were preserved with 4 °C in College of Landscape Architecture, Northeast Forestry University (Harbin, Heilongjiang). Lay the soaked seeds flat on a petri dish with wet filter paper at low density, and seeds germinated in two days. The seedlings were cultivated in a growth chamber at a temperature of 25 ± 2 °C with a light/dark cycle of 16/8 h and 60%–70% relative humidity for vegetative growth (Wang et al., 2022). At one month of age, the fourth to sixth fully expanded leaves beneath the apex were sprayed with 100 µM salicylic acid (SA), 50 µM methyl jansmonate (MeJA), 100 µM gibberellin A3 (GA3), 100 mM abscisic acid (ABA) and 0.5 g/L ethylene (ETH) hormone. The roots of seedings were soaked in 200 mmol L−1 NaCl and 20% polyethylene glycol (PEG) 6,000 to simulate salty and drought environment. Leaves were sampled followed by 0, 3, 6, 12, 24 and 48 h and immediately stored at −80 °C in preparation for subsequent experiment (Li et al., 2013; Liu et al., 2021a; Wang et al., 2021). The leaf samples of each treatment repeated three times and sprayed with sterile water as the control.

Identification and analysis of SBP-box genes in C. nankingense

The related genome data of C. nankingense was downloaded from chrysanthemum genome database (http://www.amwayabrc.com/zh-cn/index.html), and the BLAST program was set up with local environment for efficient sequence alignments. Protein sequences, coding sequences and genome data of Arabidopsis, rice and Artemisia annua were obtained from website (https://www.A.thaliana.org/index.jsp), (https://rapdb.dna.affrc.go.jp/download/irgsp1.html) and NCBI (https://www.ncbi.nlm.nih.gov/genome/). AtSPL and OsSPL protein sequences were used to identify SBP gene family members C. nankingense with sequence alignments (E-value ≤1e−5) in localized BLAST program. Subsequently, NCBI and Pfam (http://www.sanger.ac.uk) were used to search with a hidden Markov model (HMM) profile of the SBP domain (Pfam ID: PF03110) with a cut-off E-value of 1 × 10−5 (Finn et al., 2014; El-Gebali et al., 2019). NCBI-CDD (http://www.ncbi.nlm.nih.gov/structure/cdd/) and SMART (https://smart.embl-heidelberg.de/) were used to confirm whether a complete SBP domain existed or not. InterProScan based on member databases, including CATHGene3D, PANTHER, PROSITE, SUPERFAMILY and InterPro were repeated to search for these proteins in case of the missing or redundancy of SBP domains (Mulder & Apweiler, 2007). The selected SBP proteins were renamed CnSBP1-CnSBP21 according to the ascending order of genomic protein IDs. The physicochemical properties of the CnSBP proteins, including relative molecular mass, isoelectric point, average hydrophilic coefficient and others were analyzed by ExPASy (https://web.expasy.org/protparam/) and subcellular localization was predicted by WoLF PSORT (https://www.genscript.com/psort.html). The secondary and tertiary structures of proteins were predicted by SOPMA (https://npsa-prabi.ibcp.fr/cgi-bin/npsa_automat.pl?page=/NPSA/npsa_sopma.html) and SWISS-MODEL (https://swissmodel.expasy.org).

Sequence alignments, phylogenetic and gene structure analysis

Multiple alignments were carried out by DNAMAN 7.0 and ClustalX1.83. Phylogenetic trees were constructed by MEGA 7 software with parameters of neighbor-joining (NJ) method, 1,000 times bootstrap replications and p-distance substitutions model with 50% cut-off partial deletion based on 69 SBP-box genes from four species, including monocotyledons (O. sativa) and dicotyledons (Arabidopsis, A. annua, and C. nankingense) (Kumar, Stecher & Tamura, 2016). The conserved motifs of CnSBP proteins were extracted from MEME website (http://meme.nbcr.net/meme/intro.html) (Bailey et al., 2006). The parameters were set as follows: number of motifs: 8; motifs width: 6-50. The conserved sequence logos were obtained through Weblogo (http://weblogo.berkeley.edu) website. The exon-intron structure of CnSBPs was extracted by TBtools software according to the genome annotation file (gff.) files (Chen et al., 2020).

Calculation of Ka/Ks values

Due to the degeneracy of codons, the difference of paralogous and orthologous gene sequences during species evolution resulted in amino acid change in the encoded protein, which was known as non-synonymous substitution (Ka), conversely, the existence of synonymous codon in same amino acid was called synonymous substitution (Ks). Software DnaSP5 was used to calculate the Ka and Ks values aiming to analyze gene duplication events (Librado & Rozas, 2009). The Ka/Ks rate of orthologous and paralogous SBP-box gene pairs between C. nankingense and Arabidopsis was used to determine the selection pressure, and the Ks value can reflect the divergence time during large-scale duplication events. Divergence time (T) was calculated with the formula T = Ks/2 λ Mya for each gene pair to estimate the date of duplication events. The approximate clock-like synonymous substitution rate (λ) was 1.5 × 10−8 substitutions synonymous/site/year in dicots (Blanc & Wolfe, 2004; Won et al., 2017).

Promoter cis-elements, protein interaction and miR156-targeted sites prediction

We extracted 2000 bp sequences from TBtools software as promoters of CnSBP genes to excavate cis-regulatory elements for further research on regulation mechanism. The cis-regulator elements were predicted by PlantCare (http://bioinformatics.psb.ugent.be/webtools/plantcare/html/) website and visualized by TBtools (Lescot et al., 2002). STRING (https://string-db.org) online website was used to conduct a preliminary prediction of the homologous proteins of CnSBPs, AtSPLs in Arabidopsis, and Cytoscan software was used to visualize the interactive network relationship. We aligned miRNA high-throughput sequencing data in C. indicum with Arabidopsis to obtain ath-miR156 mature sequences from miRBase (https://www.mirbase.org/) and searched ath-miR156-targeted sites in psRNATarget (http://plantgrn.noble.org/v1_psRNATarget) (Dai, Zhuang & Zhao, 2018).

Expression profiles of CnSBP genes

For increasing insights into potential functions of CnSBPs, we analyzed the tissue-specific expression patterns of 21 CnSBP genes. RNA-seq data of 6 various plant tissues and organs (leaves (L), stems (S), roots (R), buds (B), ligulate flowers (LF) and tubular flowers (TF)) were downloaded from C. nankingense genome database (http://www.amwayabrc.com/zh-cn/download.htm). The expression data was extracted by transcripts per kilobase of exon model per million (TPM) mapped reads using TBtools software. The expression levels of 21 CnSBPs were showed by TBtools in the form of heatmaps with parameters of normalized scale method and log scale.

Quantitative real-time PCR analysis

Total RNA was extracted from the frozen samples using Plant RNA Extract Kit R6827 (Omega Bio-Tek, Guangzhou). Single-strand cDNA was synthesized from total RNA using ReverTra Ace® qPCR RT Master Mix (TOYOBO, Japan). Quantitative Real-time PCR was conducted with the UltraSYBR Mixture (Low ROX) (CWBIO, Beijing). The sequences of specific primers were listed in Table S1. All groups of qRT-PCR experiments were performed with three biological duplications, and gene CmEF1 α (GenBank Accession No. KF305681) was determined for reference gene (Zhu et al., 2020). The relative expression levels were calculated with the 2−ΔΔCt method (Pfaffl, 2001).

Results

Identification and characteristics of SBP-box family genes in C. nankingense

We preliminarily obtained 28 CnSBP genes from BLAST sequence alignments and HMMER with a profile Hidden Markov Model (pHMM) of the SBP domain (PF03110). However, seven of them (CHR00008556, CHR00054349, CHR00065414, CHR00077268, CHR00077269, CHR00078717, CHR00084913) were excluded from SBP-box family in chrysanthemum for further analysis in SMART and NCBI-CDD database due to their incomplete or redundant SBP domains. Analysis of PfamScan and InterProScan based on different member databases also confirmed complete SBP domains of the filtered CnSBP proteins. Eventually, 21 CnSBP genes were determined in C. nankingense genome, and we renamed CnSBP1 to CnSBP21 based on ascending order of genomic gene IDs.

The amino acid length (aa), relative molecular weight (MW), isoelectric point (PI) and average hydrophilic coefficient (GRAVY) of 21 CnSBP proteins were summarized in Table 1. The amino acid length was ranged from 142 to 954 aa and the molecular weight were in a range of 116447.45–106321.94 Kd. The 21 CnSBP proteins were mostly basic amino acids and unstable proteins, due to the above 7.0 isoelectric point and over 40 the instability coefficient. It indicated that all the CnSBP proteins were hydrophobic due to the negative value of GRAVY except CnSBP14 which was hydrophilic protein. Subcellular localization results showed 19 CnSBP proteins were predictably located in the nucleus but both of CnSBP3 and CnSBP14 were mainly located in endoplasmic reticulum, meaning additional functions may exist in CnSBP3 and CnSBP14. All of 21 CnSBP proteins possessed major secondary and tertiary structures including α-helix, β-helix, random coli and extended strand but the proportion of each structure was distinct (Table S2).

Table 1 Information on the SBP-box family genes in C. nankingense.

Gene name	Gene ID a	Protein physical and chemical properties	SBP domain location	Homologue of AtSPL / OsSPL	Exons	Subcellular localization prediction	
		Length b	MW (kd) c	PI d	GRAVY e					
CnSBP1	CHR00007823	170	19226.52	9.44	−1.082	59-133	SPL4/5/OsSPL7	2	Nuclear	
CnSBP2	CHR00009123	163	18557.94	9.08	−0.862	43-110		3	Nuclear	
CnSBP3	CHR00009124	258	28789.47	7.47	−0.119	7-74		6	Cytoplasm	
CnSBP4	CHR00010885	496	55372.13	6.39	−0.549	110-184	SPL3/6	4	Nuclear	
CnSBP5	CHR00016731	416	45571.83	6.06	−0.643	140-214	OsSPL3/12	4	Nuclear	
CnSBP6	CHR00023257	301	33148.32	9.80	−0.590	32-106	SPL13A/B	3	Nuclear	
CnSBP7	CHR00026823	302	34552.67	8.60	−1.048	195-269	SPL8	3	Nuclear	
CnSBP8	CHR00027408	954	106321.9	6.16	−0.437	145-219	SPL1/12/OsSPL6	11	Nuclear	
CnSBP9	CHR00030302	291	32661.65	9.53	−0.738	24-98	SPL3	3	Nuclear	
CnSBP10	CHR00032503	277	31611.59	9.53	−0.725	38-112	OsSPL3/12	3	Nuclear	
CnSBP11	CHR00032581	393	43423.35	9.21	−0.664	152-226	OsSPL3/12	3	Nuclear	
CnSBP12	CHR00053072	233	26179.04	6.13	−0.601	25-99	SPL13A/B	3	Nuclear	
CnSBP13	CHR00053073	197	22230.78	6.85	−0.779	20-94	SPL13A/B	3	Nuclear	
CnSBP14	CHR00057355	920	102412.1	6.19	0.018	114-188	SPL7/OsSPL9	13	Cytoplasm	
CnSBP15	CHR00058779	210	24362.60	9.86	−1.237	127-201	OsSPL7	3	Nuclear	
CnSBP16	CHR00062917	148	16447.45	9.30	−0.968	57-131	SPL4/5/OsSPL7	2	Nuclear	
CnSBP17	CHR00063016	310	34975.01	8.87	−0.720	75-149	SPL3	3	Nuclear	
CnSBP18	CHR00068589	395	43883.86	8.57	−0.770	93-167	SPL13A/B	3	Nuclear	
CnSBP19	CHR00069886	197	22211.74	6.58	−0.814	20-94	SPL13A/B	3	Nuclear	
CnSBP20	CHR00075690	428	48515.91	6.93	−0.658	150-203		3	Nuclear	
CnSBP21	CHR00083541	142	16626.62	9.28	−1.231	58-132	OsSPL7	2	Nuclear	
Notes.

a Gene ID was corresponded to the annotation provided from C. nankingense genome database.

b The amino acid length of CnSBP protein.

c Molecular weight of CnSBP protein.

d Isoelectric point of CnSBP protein.

e Grand average of hydropathicity of CnSBP protein.

Sequence alignments and phylogenetic analyses

The conserved domain sequences of 21 CnSBP proteins were showed in Table S3. As shown in Fig. 1, 21 CnSBP proteins all have an intact SBP conserved domain (SBP-DBD) which was generally composed of 72-80 amino acid residues. The SBP domain contained three features, the two zinc finger-like structures (Zn1 and Zn2) and a nuclear localization signal region (NLS). CysCysCysHis (C3H) was Zn1 structure for all members except CnSBP14 with another Zn1-like structure CysCysCysCys (C4) which was consistent with AtSPL7 in Arabidopsis. While CysCysHisCys (C2HC), the Zn2 structure existed in 18 CnSBP proteins, with the exception of CnSBP2, CnSBP3 and CnSBP12 which lacked part of the C2HC structure. Similar to Arabidopsis, the C-terminus of SBP domain in CnSBP proteins owned highly conserved NLS region consisting of a large number of basic amino acid residues. The NLS region shared partial sequence with Zn2 structure and specifically identified GTAC motif that may play an important role in regulating the accurate binding of SBP proteins to target DNA sequence and locating in nucleus (Fig. 1) (Birkenbihl et al., 2005; Riese et al., 2007).

Figure 1 SBP domain alignments of 21 CnSBP proteins.

(A) Sequence logo of the SBP domain of CnSBP proteins. The height of the letters within each stack represents the relative frequency of the corresponding amino acids. (B) Multiple alignments of the SBP domain in 21 CnSBPs were performed by DNAMAN 7.0 software. Two zinc-finger structures (Zn1 and Zn2) and a nuclear localization signal region (NLS) were marked.

According to the results, 69 SBP-box genes were clustered into eight groups (GI - GVIII) (Fig. 2). The 21 CnSBPs were distributed in all eight groups and the largest group (GVIII) contained seven CnSBPs accounted for 33.3% of the total CnSBPs, whereas GII, GIII, GIV and GVI contained only one CnSBP member. The phylogenetic tree showed that there were 4 groups of paralogous genes in C.nankingense, CnSBP2/CnSBP3, CnSBP1/CnSBP16, CnSBP12/13/19 and CnSBP9/CnSBP17, meanwhile, 10 groups of orthologous genes were found in Arabidopsis and A. annua. It was worth noting that most CnSBPs were highly homologous with AaSBPs due to close evolutionary relationships in Asteraceae species. Apart from GII, the remaining groups contained CnSBP and AtSPL gene family members. It was speculated that the CnSBP genes have undergone multiple gene replication events from the same ancestral gene and distinct patterns of differentiation occurred among many family members after the separation of each lineage.

Figure 2 Phylogenetic tree of SBP-box family proteins from chrysanthemum and other species.

The neighbor-joining (NJ) method was used to construct phylogenetic tree containing 17 Arabidopsis (AtSPL), 19 rice (LOC_OsSPL), 12 A. annua (AanSBP) and 21 C. nankingense (CnSBP) proteins. The eight subgroups were colored differently.

Motif composition and gene structures analysis of CnSBPs

The typical evolutionary blots and biological functions of TF families were linked with the intron/exon structure, therefore, we analyzed the structural characteristics between 21 CnSBP genes and 17 AtSPL genes (using the accession number in Arabidopsis) (Fig. 3A). The results revealed that CnSBP8 contained additional gene and motif structures with low complexity sequence repeats regarded as the ankyrin repeat domain (ANK-domain). The protein-protein interaction in ANK-domain mediated diverse and complex biological functions in CnSBP genes.

Figure 3 Phylogenetic tree, gene structures and motif distribution of the AtSPL and CnSBP genes.

(A) The exon-intron structures of the CnSBPs; exons (CDS) and introns were indicated by yellow boxes and black lines, and blue boxes represented the non-coding regions (UTRs). (B) Conserved motifs of the CnSBP proteins; boxes with different colors and positions represented different structural motifs. (C) The sequence logos of motif 1 and motif 2 were visualized by WebLogo online website.

The intron–exon structures indicated that different CnSBP genes were diverse, while the same subgroup genes usually possessed similar intron–exon structures, for instance, CnSBP12/13/18/19 owned three exons in GVIII (Fig. 2, Fig. 3A). Statistical analyses showed that most CnSBP genes contained 2-4 exons, but CnSBP3, CnSBP8 and CnSBP14 contained 6, 11 and 13, respectively (Fig. 3A). Most members of gene family with shared motifs likely to be an indispensable part to implement important functions or structure compositions. It is particularly critical to excavate new members of gene families by features of conserved motifs. From Fig. 3B, we selected eight motifs within AtSPL and CnSBP proteins and the sequence logos were showed in Fig. S1. It showed that most CnSBP proteins possessed three to six motifs and motif 1, 2 almost simultaneously existed in all CnSBP proteins apart from CnSBP2 and CnSBP3. According to the gene and protein structures, 38 genes were divided into four groups (GA-GD). Members of GD owned two or four extra motifs, which hinted relative specific structures and functions in GD genes. GC members didn’t share any other motifs except motif 1 and 2 (Fig. 3B). In order to display the detailed information of the motifs intuitively, the motif 1 and 2 sequence logos were showed in Fig. 3C. On a basis of sequence alignments and domain analysis in above, it was clear that motif 2 corresponded to Zn1 and partial Zn2 finger-like domain, meantime motif 1 contained the complete NLS region (Figs. 3B, 3C). The biological functions of other motifs remained unknown, so it could predicted that some CnSBP proteins had unidentified functions.

Gene duplication and evolution analysis of CnSBPs

10 paralogous gene pairs (Cn-Cn) in C. nankingense genome and 6 orthologous gene pairs (Cn-At) between the CnSBP and AtSPL genes were identified with BLASTn and ClustalX. All of the paralogous and orthologous pairs were listed in Table S4. For every homologous gene pair, we calculated Ka, Ks and Ka/Ks values to explore evolutionary selection pressure and investigate the divergence of CnSBPs (Table S4). Furthermore, the frequency distributions of the Ks and Ka/Ks values for the homologous gene pairs from C. nankingense and Arabidopsis were calculated (Fig. 4). The frequency distribution of Ks values for the paralogous pairs in C. nankingense averaged ∼0.3 (Fig. 4A), indicating that a large-scale duplication event occurred in SBP-box gene family in C. nankingense approximately 10 million years ago (Mya). Recent research has suggested that the most recent WGD event in C. nankingense occurred ∼5.8 Mya, which was a persuasive evidence that the duplicate event of the SBP-box genes occurred earlier than whole-genome WGD event. Also, for the At-Cn orthologous pairs, the average value at ∼0.72 estimated that the divergence time of the SBP-box genes was 24 Mya (Fig. 4B). Significantly, the Ka/Ks peaks in the Cn-Cn were distributed between 0.5−0.6 (Fig. 4C), while the Ka/Ks in Cn-At were 0.7−0.8 (Fig. 4D). On the basis of the values of Ka/Ks, it reflected that the SBP-box genes subjected to purification selection (Ka/Ks<1) for homologous gene pairs in Cn-Cn as well as Cn-At, and tended to eliminate harmful mutations in the population.

Figure 4 The distribution of the Ks and Ka/Ks values of the paralogous CnSBP gene pairs (Cn-Cn) and orthologous CnSBP and AtSPL gene pairs (Cn-At).

(A, C) Distribution of Ks and Ka/Ks values were obtained from paralogous gene pairs (Cn-Cn) in C. nankingense genome. (B, D) Distribution of Ks and Ka/Ks values were obtained from orthologous gene pairs (Cn-At) between C. nankingense and Arabidopsis genome.

Analysis of Cis-regulatory elements in the promoter regions of CnSBPs

The distributions and descriptions of critical cis-elements corresponding functions of CnSBP gene promoters were showed in Fig. 5A and Table S5. Light-responsiveness regulatory elements, including AE-box, 3-AF1, ACE, Box 4, G-box and others were distributed in most CnSBPs promoter regions (Fig. 5B). Besides, stress regulatory elements GC-motif, MBS, LTR, ARE, TC-rich and WUN-motif, separately in response to anoxic specific inducibility, drought-inducibility, low-temperature responsiveness, anaerobic induction, defense and stress responsiveness and wound responsiveness were respectively identified in 1, 11, 7, 17, 5 and 9 CnSBP genes. Likewise, 52 ARE elements occupied the major proportion of stress-responsive elements (Fig. 5B), providing an insight that CnSBPs may involve in anaerobic induction.

Figure 5 Cis-elements analysis of CnSBP genes promoters.

(A) The 2000 bp sequences upstream from the transcription start site were extracted. Different colored boxes represented different cis-regulator elements. (B) The total number of cis-regulator elements of 21 CnSBP genes related to hormone, light, stress and growth responsiveness. (C) Various types of hormone-responsiveness cis-elements accounted for the proportion of total hormone-responsiveness cis-elements in CnSBP genes.

83 abscisic acid response elements (ABRE), 58 MeJA-responsive elements (CGTCA motif and TGACG-motif), 14 salicylic acid response elements (TCA-element), 10 auxin-responsive elements (TGA-element and AuxRR-core) and 10 gibberellin-responsive elements (GARE-motif, TATC-box, and P-box) were identified (Fig. 5B). The percentage of various hormone-responsive elements were showed in Fig. 5C. It was worth noting that all of the CnSBP promoter regions contained at least one hormone-responsive elements. CnSBP4 and CnSBP5 only owned ABA-responsive elements and CnSBP12 owned MeJA-responsive elements (Fig. 5C). Different types and numbers of hormone-responsive elements provided sufficient bases that specific CnSBP genes may respond to exogenous hormones and ulteriorly involve in abiotic stresses.

MiR156-targeted sites prediction of CnSBPs

Target sites of miR156 in plants with close relationship tend to conserved in evolution. Due to lack of miRNA sequencing of C. nankingense, we used five mature miR156 family members (Ath-miR156i/j/e/a-5p/f-5p) in Arabidopsis to predict the miR156-targeted sites in 21 CnSBP genes initially. Multiple sequence alignments of the CnSBP genes and reverse complement sequences of Ath-miR156 showed that 11 CnSBPs contained highly consistent sequences with Ath-miR156 binding sites with no more one to three mismatches (Fig. 6). It suggested that cna-miR156 may specifically target these genes in C. nankingense. These putative miR156 response elements (MREs) of CnSBP genes were located downstream of the SBP-box in the coding region of genes in groups GV (CnSBP4/9/17), GVI (CnSBP6/13/18/19) and GVIII (CnSBP5/10/11/20).

Figure 6 Alignment of miR156-targeted sites complementary sequences within CnSBP genes and ath-miR156 in C. nankingense and Arabidopsis.

Interaction prediction of CnSBP proteins

On the basis of homologous proteins of 21 CnSBP in Arabidopsis, it may have functional similarities to further predict the protein functions of CnSBPs. AtSPL proteins in Arabidopsis converged intricate protein-interaction regulation network and SPL5, SPL7 and SPL8 were pivotal central regulators related to complex functions (Fig. 7A). For example, homologous protein of CnSBP1, AtSPL5 converged many interacting proteins, such as SNZ, SMZ, AGL8, AGL20 and TOE2 (Fig. 7B). SNZ and SMZ were AP2-like ethylene-responsive transcription factor and might be involved in the regulation of gene expression by stress factors and by components of stress transduction pathways. It provided an insight that CnSBP1 might play critical regulation roles in hormone signal transduced pathway and abiotic stresses. AtSPL7 (homologous protein of CnSBP13 and CnSBP20) interacted with SIZ1 which involved in the regulation of plant growth, drought responses, freezing tolerance and salicylic acid (SA) accumulation (Fig. 7C). Besides, SPL8 interacted with AGL8, AGL18, AGL20 and AP1 (MAD-box gene family) (Fig. 7D). AGL8 involved in developmental growth in morphogenesis and positively regulated flower development, on the contrary, AGL18 had negatively regulation of flowering. And AGL20 regulated flowering and inflorescence meristem identity and responded to gibberellin.

Figure 7 Potential protein–protein interaction network of CnSBPs.

(A) CnSBP1 and CnSBP16 were clustered with homologous SPL5 protein in A. thaliana. (B) CnSBP14 was clustered as homologous SPL7 protein in A. thaliana. (C) CnSBP7 was clustered as homologous SPL8 protein in A. thaliana.

Tissue-specific expression profiles of CnSBP genes

The patterns of gene tissue-specific expression often have a correlation with its encoded protein function. Publicly available transcriptome data of six tissues (root, stem, leaf, bud, ligulate flower and tubular flower) showed transcript levels and cluster analysis (G a-e) of 21 CnSBP genes (Fig. 8, File S1). It showed that more than two-thirds of CnSBP genes significantly expressed in floral tissues by comparison with one-third expressed in root, stem and leaf tissues. Among these, CnSBP3 and CnSBP7 only showed a high expression level in the stage of flower development, and CnSBP4 evidently expressed in roots. Overall, eight CnSBP genes (CnSBP5/9/11/14/17/18) in group e shown constitutive expression patterns in all six tissues/organs, while group c and d showed lower expression levels across the nutritive organs than reproduction organs. CnSBP9/14/17/18 have relatively high expression levels in leaf and CnSBP8 and CnSBP21 significantly expressed in all tissues. With regard to tissue-specific expression patterns, the majority of miR156-targeted CnSBP genes showed higher expression levels in floral tissues instead of non-targeted CnSBP genes. For example, miR156-targeted CnSBP5/9/11/17/18 (members of G e) genes significantly expressed in all tissues, and miR156-targeted CnSBP13/19 genes tended to exhibit higher transcript levels in floral tissues. In terms of CnSBP genes in group a, CnSBP8 expressed ultrahigh transcript levels in all six tissues, and CnSBP21 similarly showed expression trend but almost no expression in roots (Fig. 8).

Figure 8 Expression profiles of CnSBP genes in six tissues and organs (buds, ligulate flowers, tubular flowers, leaves, roots and stems).

Orange and blue indicated high and low expression levels by TPM values in transcript.

Expression profiles of CnSBP genes under plant hormone and abiotic stresses

The expression patterns of CnSBP genes under plant hormones treatments were examined to the responsive profiles and functions of CnSBPs by qRT-PCR (Fig. 9). The raw datas of 21 CnSBP genes with ABA, GA, MeJA, SA and ETH treatments were placed in (File S2, S3, S4, S5 and S6). Oligonucleotide primers of 21 CnSBP genes and actin gene sequences were listed in Table S1.

Figure 9 Expression levels of CnSBP genes in leaves under hormone treatments by qRT-PCR.

The Y-axis indicated the relative expression level; X-axis (0, 3, 6, 12, 24 and 48 h) indicated hours post hormone treatments. Different colors represented different hormone treatments (ABA, GA3, MeJA, SA and ETH). The standard errors were plotted using vertical lines. The experiments in all panels were repeated three times until convincing results. Bars with different lowercase letters were calculated by one-way ANOVA in SPSS 23.

Majority of the CnSBP genes expression could be induced or inhibited response to GA3 phytohormones. CnSBP5, CnSBP8, CnSBP13 and CnSBP19 were evidently upregulated by nearly 2.47-, 3.24-, 2.81- and 3.18- fold during 12 h treatment, among these, CnSBP3/5/13/14/15/19 increased in expression at all stages, but CnSBP4/8/9 were induced to a peak at 12 h and had a downward trend from 24 h to 48 h (Fig. 9). Under ABA treatment, most CnSBP genes downregulated from 3 h to 6 h, but gradually upregulated during the follow-up periods or reached a maximum peak at 12 h. All the remaining CnSBP genes displayed a inconspicuous expression fluctuation, for instance, CnSBP13/14/19 increased after slight drop in expression levels. CnSBP2/3/7/12 showed an obvious upward trend in response to MeJA before 12 h, CnSBP4/7/12 performed an obvious decrease in transcript levels from 24 h to 48 h. CnSBP14/17/18/21 exhibited slightly decreases along with various point of time. Following SA treatment, most CnSBP genes presented a decreased trend, except CnSBP9/17 prominently increased. Additionally, other CnSBP genes displayed slight up- and downregulated fluctuations during processing of SA. Finally, it occurred that the expressions of most CnSBP gene upregulated at apex of 12 h or 24 h, but descended from 24 h to 48 h response toETH treatment. In general, CnSBP1/7/11/14/16/18 significantly upregulated during the whole process, and tandem duplicated genes (CnSBP1/16 and CnSBP9/17) showed similar expression pattern throughout various hormone treatments (Fig. 9). We also observed that the same subgroup CnSBPs showed a distinct expression trend, such as CnSBP10, CnSBP11 and CnSBP20 in GVII (Figs. 2, 9). It suggested that specific CnSBP genes might play multiple roles in hormone signal pathway and activate the adaptive regulatory responses in plants and participated in the regulations of abiotic stresses.

In order to investigate the mechanism of resisting stresses dependent on hormone signal pathway, the expression profiles and raw data of 21 CnSBP genes in response to salt and drought stresses were examined by qRT-PCR (Fig. 10, File S7 and S8). It showed that most CnSBP genes more or less affected by salt and drought treatments, implying that CnSBP genes may play a pivotal role in response to abiotic stresses. In detail, CnSBP5/12/13 (2.35, 1.50 and 2.05 fold), CnSBP2/7/20 (2.14, 2.29 and 1.43 fold) and CnSBP1/3/6/11/15/16/17/21 (1.50, 1.91, 1.63, 1.61, 1.66, 1.62, 1.54 and 2.47 fold compared to 0h) were significantly upregulated by salt stress at early (0 h–6 h), medium (6 h–12 h) and late (12h-48h) responsive periods, respectively (Fig. 10). It exhibited expression trend that firstly increased and then decreased with the passing of time in CnSBP5/7/8/12/13/20. Under drought treatment, CnSBP12/13/15/18 performed descending expression levels (0.63, 0.47, 0.69 and 0.49 fold at 48 h) during the whole periods of time; CnSBP7/9//10/14/17/19 showed initially increasing then decreasing trend (Fig. 10). Interestingly, the vast majority of CnSBP genes had no large multiple differentially induced or downregulated under salt and drought stresses. In general, specific CnSBP genes showed co-expression levels in hormone signaling and abiotic stresses, indicating that complex regulatory network covered the processes of plant responsing to stresses and hormone signal transduction.

Figure 10 Expression levels of CnSBP genes in leaves under abiotic stresses by qRT-PCR.

The Y-axis indicated the relative expression level; X-axis (0, 3, 6, 12, 24 and 48 h) indicated hours post abiotic stresses. Different colors represented different abiotic stresses (salt and drought). The standard errors were plotted using vertical lines. The experiments in all panels were repeated three times until convincing results. Bars with different lowercase letters were calculated by one-way ANOVA in SPSS 23.

Discussion

Traditional Chinese flowers, chrysanthemum, is famous for petal colors and floral morphological characteristics. Owning to the nature diploid and progenitor genome, C. nankingense, a close relative of C. morifolium, has been considered as a convenient genomic model to research in chrysanthemum (Song et al., 2018). Chrysanthemum is susceptible to several abiotic stresses including salt and drought, which has adverse impacts on growth, morphology development, quality, thus leading to serious economic losses. SBP-box gene family, a class of plant-specific transcription factor, evolved before the divergence between green algae and the ancestor of land plants, proving that widely involved in life processes such as plant growth, floral development, flowering, fruit ripening, biotic and abiotic stresses and hormone signaling pathway. Identification and expression patterns analysis have discussed on 12 CmSPL genes in response to hormones and stresses on the basis of C. morifolium transcriptomic data (Song et al., 2016). In this study, we identified 21 CnSBP family genes from C. nankingense genome and provided new insights for comprehensive understanding of the SBP-box genes in non-model plants (Fig. 1). Compared with crops, cotton (83 GhSBPs), maize (42 ZmSBPs), oilseed rape (58 BnaSBP) and wheat (50 TaSBPs), C. nankingense contained much less SBP-box genes (Zhang et al., 2015; Cheng et al., 2016; Peng et al., 2019; Li et al., 2020), but resembled the model plant Arabidopsis (17 AtSPLs), flowering plants petunia (21 PhSPLs), Prunus persica (17 PpSPLs), Prunus mume (17 PmSPLs) and Rosa rugosa (17 RcSPLs), indicating that the SBP-box family genes endowed with more diversified and complicated functions with species specificity. It could be a consequence of the divergence of flowering responsive functions in SBP-box genes.

Physicochemical properties of proteins showed that 21 CnSBP were almost basic amino acids, unstable and hydrophobic proteins. The predictions of secondary and tertiary structures concluded that all CnSBP proteins own similar structures except for subtle diversities, which may lead to various functions (Table S2). Studying the conserved domains of CnSBP genes was conducive to highlight the cognition of the SBP-box structure. All of the CnSBP proteins contained a complete SBP domain consisting of two zinc finger-like structures (Zn1 and Zn2) and a nuclear localization signal region (NLS) analyzed by Pfam and InterProScan with member databases (Fig. 1). It was unique that the Zn2 and NLS regions shared the common four amino acid residues (KRSC). Unlike other zinc finger structures owned a staggered binding mode, Zn2+ and NLS region were necessary for binding to cis-elements to the promoters of nuclear genes. Moreover, CnSBP8 possessed an extra ANK- domain in the C-terminal (742-843 aa) of protein, which had a bearing on protein-protein interactions in plant cells (Lee et al., 2016). It was clear that the ANK-domain corresponded to motif 4 and motif 8 and encoded correlative exon sequences (Fig. 3). Likewise, CsSBP12 and CsSBP10b in sweet orange and AtSPL14 in Arabidopsis with the same ANK- domain were separately in sensitivity to pathogen Diaporthe citri and fungal toxin Fumonisin B1 (FB1) (Stone et al., 2005; Song et al., 2021). It perhaps indicates that CnSBP8 plays a pivotal role in biotic stresses such as pathogen fungal infection.

Based on phylogenetic tree and gene structure analysis, 21 CnSBPs were clustered into eight groups (GI - GVIII) from four species and exhibited closer homology to Arabidopsis (17 AtSPLs) and A. annua (12 AaSBPs) rather than rice (OsSPLs) suggesting that conservative evolution and common ancestor shared in Compositae and dicots plants away from the lineage leading to monocots (Fig. 2). The exon-intron structures and motif analysis also provided significant determinants to cluster phylogenetic tree to a point. The same group always shared similar structures, such as the members of GVI, CnSBP12/13/18/19 contained motif 1/2/6/7 and three exon distributions (Fig. 3B), indicating that the evolution and gene structures may be interrelated. Besides, separate branch members in GI and GIII owned more complex motifs and gene structures implying that CnSBP8 and CnSBP14 may perform additional functions and independent evolution similar to CsSBP11 in sweet orange (Fig. 3) (Song et al., 2021). Intriguingly, on the basis of amino acid sequence alignments, it seemed that CnSBP8 owned a comparable AHA-like domain outside the N-terminal and a IRPGC motif outside the C-terminal of the SBP-domain, which was characteristic of many transcriptional activation domains consistent with CRR1 in C. reinhardtii (Fig. S2) (Riese et al., 2007). The sequence logos of AHA-like and IRPGC motif were showed in (Fig. S4). The same structures were also found in AtSPLs and OsSPLs clustered with CnSBP8 in group with complex motifs and intron-exon hinting that unknown functions combined with gene structures (Fig. 3, Figs. S2, S3). Furthermore, there was a conserved IRPGC motif existed in downstream of the SBP domain, which was also found in CRR1 in C. reinhardtii (Figs. S3, S4) (Kropat et al., 2005). It was reported that SPL7 (homologous gene of CnSBP8) played a central role in regulating of Cu2+ and transmembrane transporter activity and SPL12 (homologous gene of CnSBP14) regulated root tip and embryonic meristem development, nitrogen metabolism and plant thermos-tolerance at reproductive stage in Arabidopsis (Chao et al., 2017; Kastoori Ramamurthy et al., 2018).

SBP-box genes had underwent duplication event leading to the formation and preservation of multiple paralogs and evolutionary branches. As evident from the phylogenetic tree and BLASTn, 4 pairs of duplicated genes (CnSBP2/3, CnSBP1/16, CnSBP9/17 and CnSBP13/19) were identified (Fig. 2) in accordance with Arabidopsis and rice, indicating that duplicate genes might result in amplified SBP-box family in C. nankingense (Yang et al., 2008). The results of homologous gene comparison for fragment duplication were highly consistent with phylogenetic tree clustering scheme of the evolutionary group (Fig. 2). To explore the macroscopic evolution model in C. nankingense, the Ka/Ks ratios for the duplicated gene pairs were estimated. Significantly, the Ka/Ks peak ratios for the Cn-Cn and Cn-At gene pairs were not difference, respectively, 0.5−0.6 and 0.7−0.8, suggesting that the CnSBP genes experienced a strong constraint and purification selection to get adaptive growth in various environment (Fig. 4). As discussed, the Ks values confirmed that the CnSBP genes approximately occurred duplication events ∼10 and ∼24 Mya ago earlier than the recent whole genome duplication (WGD) event between C. nankingense and Arabidopsis, indicating that the SBP-box gene family experienced an earlier divergence than the separation of the two most recent species (Fig. 4). In accordance with moso bamboo, SBP-box genes family occurred a positive and neutral selection in CnSBPs and PeSPLs (Pan et al., 2017). Additionally, increasing chromosomal localization of SBP-box genes in C. nankingense may contribute to the deeper understanding of homology and evolutionary relationship.

Remarkably, recent researches found that 11 out of 17 AtSPL and 11 out of 19 OsSPL genes were targeted by miR156/157, here, the miRNA response element (MRE) with speculative miR156/157-targeted sites was located downstream of the SBP domain and part of the last exon (Fig. S3) (Xie, Wu & Xiong, 2006; Riese et al., 2007; Xing et al., 2010). In this study, 11 out of 21 miR156-targeted CnSBP genes were calculated and all clustered in clades of GV, GVI and GVII with common conserved region in motif 6 (Figs. 2, 3B). It was consistent with previous researches that proved 11, 6, 12 and 19 miR156-targeted SBP-box genes in P. mume, melons, grape and walnut. It may be a major determinant of miR156-targeted SBP-box genes to carry out distinctive and significant functions with miRNA-SBP/SPLs modules in evolution. The miR156b targeted two paralogous genes, SPL9 and SPL15, controlled shoot maturation and the temporal initiation of rosette leaves (Schwarz et al., 2008). TaSPL3/17 interacted with DWARF53 to reveal potential association in SL signaling pathways during bread wheat tiller and spikelet development by miR156 targeted SPL genes (Liu et al., 2017a). Besides, miR156-targeted CnSBP5/10/11/13/17/18/19 highly expressed in floral organ (Fig. 7), demonstrating that CnSBP genes, as well as their regulators miR156 remained to regulate flower morphological characteristics. It would be relevant that SPL3 (clustered with CnSBP9 and CnSBP17) regulated by miR156 to integrate endogenous signals into flowering pathway (Gandikota et al., 2007).

Protein-interaction network conducted a preliminary prediction of CnSBP proteins. Through interacted relationships, it provided insights that CnSBP5 might play a critical regulation role in hormone signal pathway and abiotic stresses. And the homologous protein of CnSBP7, AtSPL8, largely involved in promoting flowering, inflorescence meristem identity and GA response. In summary, CnSBP proteins may combine with correlative genes involved in biotic and abiotic stresses, phytohormone pathway as well as growth and development in plants.

Tissue-specific expression analysis showed that most CnSBP genes highly expressed in floral organs possibly due that SBP proteins interacted with the SQUAMOSA (a MADS-box) promoter, a floral meristem gene correlated with the origin and evolution of reproductive organs such as flowers and ovules. Eight members (CnSBP5/8/9/11/14/17/21) showed high levels expression in all tissues regarded as significant regulatory factors in plant growth process (Fig. 8). In group b, six CnSBP genes exhibited lower expression levels in six tissues compared with other members. Interestingly, paralogous genes CnSBP2 and CnSBP3, differentially performed expression levels in floral organs, it perhaps associated that the expanded CnSBP genes occurred functional divergence resulting in novel biological functions. In group c, same subgroup members CnSBP13 and CnSBP19 expressed in floral organs and leaves. Likely, homologous gene AtSPL13A/B participated in the formation of leaf shape and reproductive stages. Furthermore, AtSPL3 (clustered with CnSBP4) regulated flowering time and activated downstream gene expression during flowering morphological development (Jung et al., 2012). OsSPL9 (clustered with CnSBP14) regulated the number and yield of grains as well as Cu accumulation and metabolism in rice, suggesting potential roles in CnSBP14 (Tang et al., 2016).

During the lengthy evolution of organisms, plants have obtained complex genic regulatory mechanisms to mitigate effects from adverse environments. Both enzymes and hormones were crucial means by which plant affected a series of physiological or biochemical changes to gain adaptive capacity to resist the stresses (Sah, Reddy & Li, 2016). In the study, a further finding was that numerous of hormone-responsive elements as well as stress-responsive elements were exhibited in CnSBP promoters, hinting that 21 CnSBP genes may have an intense response to hormone signal and abiotic stresses (Fig. 5). Therefore, in line with the ideas that co-expression of genes in response to abiotic stresses and exogenous induction were considered as candidate genes to involve in regulation. Consequently, we researched the expression profiles of the CnSBP genes under ABA, SA, MeJA, GA3 and Eth hormone treatments. Exogenous spraying induction can not only activate the expression of defense-related genes, but also interconnect hormonal signal network with defense responses. Expression analyses showed that 11 out of 21 members were significantly induced by ABA treatment at 12 h with a high proportion of ABA-responsive elements (Figs. 5, 9). ABA, regarded as a positive signal of stress, can improve plant tolerance to variable environment by inducing the production of H2O2 and establish ROS balance (Mittler & Blumwald, 2015). Research showed OsSPL7 (orthologous gene of CnSBP15 and CnSBP21) in rice was proved to play a critical role in ROS balance in response to biotic and abiotic stresses (Hoang et al., 2019), indicating that CnSBP15 and CnSBP21 may involve in stress responses via ABA signaling pathway. In our study, CnSBP15 and CnSBP21 significantly induced by ABA and salt treatments. 10 out of 21 members were markedly induced by GA3 treatment with poly-type GA-responsive elements, which may represent more complex expression and regulation patterns (Fig. 5, Fig. 9). An example was AtSPL3, clustered with CnSBP4, integrated photoperiod and GA signals to regulate flowering via SOC1-SPL module (Jung et al., 2012). In chinese chestnut, CmSPL6/CmSPL9/CmSPL16 highly and CmmiR156 lowly expressed during flowering development by exogenous GA3 spraying (Chen et al., 2019). Moreover, we revealed CnSBP6/9/17 prominently induced by SA and salt treatments at 24 h and 48 h time points, and CnSBP6/7/12 induced by MeJA but downregulated by drought treatment at 48 h time point (Fig. 9). It confirmed that plants induced trans-activating factors to activate promoters of defense genes related to SA pathway to improve resistance in Arabidopsis (Dong, 1998). In grape, VvSBP17 was upregulated response to SA and pathogen infection treatment which was the same as homologous gene, AtSPL14, in sensitivity to fumonisin B1 (FB1) (Hou et al., 2013). Previous studies have proved miR156-resistant SPL13 involved in ethylene biosynthesis by upregulating the expression of ACC oxidase gene in accordance with the same subgroup members, CnSBP12/13/18/19, with inductive expression patterns. Similarly, 12 MdSPLs upregulated and one MdSPL downregulated in apple by exogenous ethylene spraying (Li et al., 2013). Bioinformatics and molecular technology were limit to explore complex functions of SBP-box genes, subsequently the physiological biochemistry of candidate SBP-box genes need to investigate in future studies to better clarify regulatory network of various environment conditions.

Although the dominant roles of SBP-box genes have been explored in processes of plant growth and development, the cross-talk analysis between various stresses and hormonal response were also worthy to discuss. SA and MeJA can active multiple defense strategies and converge complex signaling networks to enhance the stress resistance capacity in plants, such as salinity stress (Qiu et al., 2014; Kim et al., 2018). In grape, the expression of VvSBP9/14/16 were downregulated expression in response to SA and MeJA and salt stress (Hou et al., 2013). CsSBP3/4/8/13 genes in tea plant (Camellia sinensis) significantly upregulated under MeJA and drought treatments (Zhang et al., 2020). DELLAs and some regulators in GA and ABA signaling pathway participate in the regulation of tolerance in response to abiotic stresses in plants. Special PeSPL genes induced by GA but inhibited by drought stress in moso bamboo (Pan et al., 2017).

In our study, most CnSBP genes exhibited various transcript levels but presence of co-expression candidate genes confirmed that SBP-box family regulated the resistance physiology of chrysanthemum via by complex stress responsive mechanism and regulation network. qRT-PCR analysis showed CnSBP1/3/5/16 were upregulated and CnSBP8/12/13/15/18 were downregulated under drought stress (Fig. 10). Among these, most CnSBP genes were prominently induced by at least one hormone accompanied by MBS (drought-inducibility) cis-elements in promoter regions. The expression levels of CnSBP1/3/9/13/15/18/21 were significant under salt treatment perhaps because they integrated with TC-rich repeats in promoter regions (Fig. 5, Fig. 10), which was regarded as defense and stress responsiveness cis-element. In rice, overexpression OsSPL10 (clustered with CnSBP7 and AtSPL8) weakened salt tolerance (Lan et al., 2019). In Alfalfa, MsamiR156-MsSPL module partially improved drought tolerance via overexpression MsamiR156 to silence MsSPL13 (Arshad et al., 2017).

In addition, lots of evidence indicated that miR156/SBP (SPL) modules regulated a variety of developmental processes and abiotic stress response in plants (Jerome Jeyakumar et al., 2020), for instance, the upgregulated expression levels of ath-miR156 inhibited targeted SPL2/9/11 genes to balance the adverse bearing on heat stress during plant growth and development (Stief et al., 2014). Besides, it was reported that MdWRKY100 gene expression was upregulated by miR156/SPL module to regulate salt tolerance in apple (Ma et al., 2021). Recently, genetic engineering means aim at enhancing tolerance of abiotic stresses by modifying miR156-targeted nodes and elucidating targeted genes will expand adaptive plants to abiotic stresses. With sequence alignments of miR156-targeted genes sites, it preliminarily cleared that specific CnSBP genes were core factors in integration of phytohormone signaling and abiotic stresses, which need to verify by further experiments in future.

Conclusions

In this study, we identified 21 SBP-box genes in C. nankingense genome and provided a comprehensive overview of SBP transcription factor family in chrysanthemum. 21 CnSBPs were classified into eight groups based on SBPs (SPLs) genes in Arabidopsis, rice and A. annua and closer homology with Arabidopsis and A. annua. Further analysis of conserved domain, motifs, gene structures, gene duplication and evolutionary supported classification results. Subsequently, we predicted physiochemical properties, secondary and tertiary structures, promoter cis-regulator elements, miR156-targeted sites and protein-protein interaction of 21 CnSBP genes. Tissue-specific expression profiles revealed that CnSBPs may play a pivotal role in floral organ growth and development. CnSBPs also responded to exogenous hormone induction and abiotic stresses. The expression patterns with same clustering results tended to be consistent. Taken together, our results helped shed light on SBP-box gene basic information in C. nankingense and provided an experimental basis on the functions of CnSBP genes in plant growth regulation. Candidate CnSBP genes should further elaborate comprehensive understanding of the co-related regulatory patterns of hormone responses and abiotic stresses. It laid a theoretical foundation for the subsequent study of miR156/SBP (SPL) modules regulation mechanism and improvement of chrysanthemum breeding.

Supplemental Information

Supplemental Information 1 Raw data for different tissues of the CnSBP genes

Each data indicates the TPM values of the CnSBP genes in transcriptome.

Click here for additional data file.

Supplemental Information 2 Raw data for ABA treatment of CnSBP genes

Each data indicates the Cq values of the CnSBP genes by qRT-PCR.

Click here for additional data file.

Supplemental Information 3 Raw data for GA3 treatment of CnSBP genes

Each data indicates the Cq values of the CnSBP genes by qRT-PCR.

Click here for additional data file.

Supplemental Information 4 Raw data for MeJA treatment of CnSBP genes

Each data indicates the Cq values of the CnSBP genes by qRT-PCR.

Click here for additional data file.

Supplemental Information 5 Raw data for SA treatment of CnSBP genes

Each data indicates the Cq values of the CnSBP genes by qRT-PCR.

Click here for additional data file.

Supplemental Information 6 Raw data for ETH treatment of CnSBP genes

Each data indicates the Cq values of the CnSBP genes by qRT-PCR.

Click here for additional data file.

Supplemental Information 7 Raw data for salt treatment of CnSBP genes

Each data indicates the Cq values of the CnSBP genes by qRT-PCR.

Click here for additional data file.

Supplemental Information 8 Raw data for drought treatment of CnSBP genes

Each data indicates the Cq values of the CnSBP genes by qRT-PCR.

Click here for additional data file.

Supplemental Information 9 Output results of CnSBP proteins with InterProScan member databases

Click here for additional data file.

Supplemental Information 10 Conserved domain analysis of preliminary identified 28 CnSBP proteins

Click here for additional data file.

Supplemental Information 11 Motif analysis of CnSBP genes in C. nankingense

Conserved motifs showed by TBtools and sequence visualized by Weblogo and MEME websites.

Click here for additional data file.

Supplemental Information 12 Multiple sequence alignments of the AtSPL, OsSPL and CnSBP proteins in group

The location of the first red box upstream of SBP domain was AHA-like motif and the third of red box downstream of SBP domain was IRPGC motif with specific conserved aa residues.

Click here for additional data file.

Supplemental Information 13 Reconstruction of the phylogenetic relationships within the SBP-box gene family in Arabidopsis, rice and C. nankingense

The conserved sequences characteristic for the different subfamilies summarized by a letter code between brackets at the end of gene names. H, AHA-like motif; S, SBP-domain; I, IRPGC-domain; A, ankyrin repeat region and M, MRE-element.

Click here for additional data file.

Supplemental Information 14 Motif sequence logos within group (Fig. S3) of eight SBP proteins

(A) Amino acid sequence logo of the AHA-like1 motif in eight CnSBPs. (B) Amino acid sequence logo of the IRPGC motif in eight CnSBPs. Each logo consisted of stacks of symbols, one stack for each position in the sequence.

Click here for additional data file.

Supplemental Information 15 The primer sequences of 21 CnSBP genes for quantitative real-time PCR

Click here for additional data file.

Supplemental Information 16 The secondary and tertiary structures of CnSBP gene family

Blue presents alpha helix; Green presents beta turn; Red presents extended strand; Pink presents random coil.

Click here for additional data file.

Supplemental Information 17 Conserved domain sequences of CnSBP proteins

Click here for additional data file.

Supplemental Information 18 Ka, Ks and Ka/Ks values calculated for homologous SBP-box gene pairs in C. nankingense and Arabidopsis.

Click here for additional data file.

Supplemental Information 19 Critical cis-regulatory elements distribution of the CnSBP genes promoters

Click here for additional data file.

Special thanks to Dr. Fadi Chen and Dr. Shilin Chen of Nanjing Agricultural University and Institute of Chinese Materia Medica for the provision of Chrysanthemum nankingense genome and transcriptome data used in this study. We are grateful to Dr. Nuananong Purente for her help in English proofreading. We would like to thank Dr. Farrukh Azeem Editor and three reviewers for their professional and helpful comments on the original manuscript.

Additional Information and Declarations

Competing Interests

Author Contributions

DNA Deposition

Data Availability

The authors declare there are no competing interests.

Ziwei Li conceived and designed the experiments, performed the experiments, prepared figures and/or tables, authored or reviewed drafts of the article, and approved the final draft.

Yujia Yang performed the experiments, prepared figures and/or tables, and approved the final draft.

Bin Chen analyzed the data, authored or reviewed drafts of the article, and approved the final draft.

Bin Xia analyzed the data, authored or reviewed drafts of the article, and approved the final draft.

Hongyao Li analyzed the data, prepared figures and/or tables, and approved the final draft.

Yunwei Zhou conceived and designed the experiments, authored or reviewed drafts of the article, and approved the final draft.

Miao He conceived and designed the experiments, authored or reviewed drafts of the article, and approved the final draft.

The following information was supplied regarding the deposition of DNA sequences:

The reference gene, CmEF1α, sequences described here are available at GenBank: KF305681.

The following information was supplied regarding data availability:

The raw data are available in the Supplemental Files.

The raw data for ABA, GA, SA, MeJA, ETH, salt and drought treatments of CnSBPs is showing the involvement of the SBP-box gene family response to hormone response and abiotic stresses in Chrysanthemum as well as the transcript expression levels in different tissues.

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
