# Peer review of "Genome-wide identification and expression analysis of SBP-box gene family reveal their involvement in hormone response and abiotic stresses in Chrysanthemum nankingense"

_PeerJ, doi:10.7717/peerj.14241_

## Round 0.1 · original submission · Major Revisions

Please address all the comments.

Reviewer 1 ·

Basic reporting

The text is clear and the literature references need to be expaned slightly.

Experimental design

The investigation was conducted rigorously.

Validity of the findings

The underling data were provided and the data were statistically sound.

Additional comments

The authors identified 21 SBP-box genes using the published whole genome sequence of Chrysanthemum nankingense. Then they analysed these genes by transcriptome data and phylogenetic based method and found that CnSBP genes may involved in plant regulation pathways. This is a routine research characterizing features of gene families from the genome. Although I do not find major flaws in the manuscript, I also do not find significant novel insights. In section Introduction, the authors used four paragraphs to introduce functions of SBP-box genes which have been well studied in other plants. However, what the authors have done is not closely-related to functions of CnSBP genes. By contrast, the authors put less emphasis on why chose Chrysanthemum as a model. I suggest the authors reduce introduction of SBP functions whereas increase the more general introduction of Chrysanthemum. For example, Chrysanthemum has many cultivars and has great medicinal and ornamental value. Many modern Chrysanthemum cultivars are of hybrid origin (Ma et al, 2020, JSE; Qi et al., 2021, JSE) et al. Finally, the authors mixed methods and discussions with section Resutls also mixed some results into Discussion. Some specific comments were listed below

Specific points:
L43-45 Need references to back it up.
L191-192 The authors noted that Ka/Ks reflects selection pressure, but did not mention it in results or discussion. I wonder why only chose Arabidopsis rather than more other species including the recent sequenced Chrysanthemum species?
L231-232 belong to section Methods
L239-242 should belong to section Discussion
L254-257 belong to section Discussion
L260-262 belong to section Methods
L277-281 belong to section Methods
L327-330 belong to section Discussion
L350-352 belong to section Methods
L361-365 belong to section Discussion
L378-385 belong to section Discussion
L388-390 belong to section Methods
L391-404 read as if Discussion
L474-476 belong to section Introduction
Figure 2 Why not include support values in the figure?
Figure 4 Not clear the parameters used to calculate Ka/Ks. What are the window size and step size? How do you deal with indels?

Reviewer 2 ·

Basic reporting

The work of Li et al. presents the initial search and characterisation of the SBPbox family in Chrysanthemum nankingense. The novelty of the work lies only in the species studied. The rest of the methodology, approaches and even the way of presenting the results mimic many previous works on SBPbox search and characterisation in other plant species (e.g. https://www.mdpi.com/2073-4425/12/11/1740/htm, /https://doi.org/10.1007/s10725-020-00677-2m, https://www.mdpi.com/1422-0067/22/16/8918/htm, among many others).
I am concerned that repeating previous studies naturalises some issues, which can certainly be improved. One of them is that, since the first bioinformatic work on the identification of genes in this family was published, only the PFAM motif PF03110 has been used as a tool for "identification" of SBPbox family members, from proteomic data. Today there are more than 14 databases of protein motifs and domains, in addition to PFAM (e.g. interpro, gene3D, Panther, etc.). Many of them can even be run simultaneously, with the same tool (e.g. Interproscan), and thus obtain information that may be missing when using PFAM alone. Thus, for the publication of this article, given that the objective is the identification and characterisation of the complete SBPbox repertoire in the species, I recommend repeating the search for these proteins by combining different tools for calling them, and on the basis of the results obtained, repeating the analyses.
I attach other comments to contribute to the improvement of the article:

Experimental design

- The link (http://blast.ncbi.nlm.nih.gov/Blast.cgi, line 160) corresponds to BLAST, and not to the FTP or NCBI site from where they downloaded genome/ proteome/ transcriptome data. Please correct
- PF03110 (line 162) is not an Accession Number, as the text indicates. It is a PFAM ID.
- Why did authors only use that ID to identify the SBPboxes. Did authors try or look for what information other databases give? Prosite, Gene3D, Interpro, Panther, etc. I think that basing all the identification work on the output of a single PFAM domain is an incomplete exploration. Repeat the analysis by extending the search to domains with other databases. You can use interproscan for this, which runs 14 databases simultaneously.
- Why did you only generate 8 MEME motifs? The text says "8 motifs found" (line 180). The significant MEME were only 8, or there are more significant motif but only the first 8 were selected and reported? If so, what were the selection criteria?
- How did you obtain or generate the SBPbox information from Arabidopsis, rice and Artemisia annua, which you used for the phylogenetic analyses?
- Since there is a report of SBP genes in a nearby species (Song 2016), it would be interesting to see if the SBPs they found here are homologous to those reported by Song. Also, it would be interesting to see the synteny of these genes in related species.

Validity of the findings

After re-analysing the data, as mentioned above, correct the "results" and "discussion" sections accordingly.

in addition:
In Figure 3, sort the legends (motifs are left unordered in legend).
I would send some figures to supplementary material, to shorten the length of the article.

Additional comments

Throughout the article
- I recommend a thorough revision of the English, by an English speaker. There are many errors that make the article difficult to read.
- I recommend that the article be corrected by a person with experience in bioinformatics. There are sentences that make no sense, or that are not really saying what they should say, for example: "the blast installation package was constructed with localization environment for efficient sequence alignments" (line 155-156).

·

Basic reporting

Overall, the paper is good for the majority part, however, there are lot grammar that needs to be improved. I have attached the work document where the grammar needs to be corrected (Please see the grammar highlighted in red).

Background was provided in a reasonably strong manner, however, more context can be added as to what might be the drawback of the current scenario/process/

Figures are not manipulated and retains it's originality. However, the tables/datasets that are attached needs legends/description of what it is trying to convey. I believe this is missing and need to be improved in great detail. For example in some of the datasets (excel files) there are mean and std deviation that are red color coded, having legends as to why these are color coded as red needs to be address/described in these tables or datasets. Also, more description regarding gene names also with references needs to be provided. Otherwise, the article is well structured.

Results are explained in great detail and I am good with the explained and results provided.

Experimental design

- Originality is organic and is retained.

- Research question is well defined and its fills in the required knowledge gap.

- The investigation conducted is rigorous and of high technical standard, nothing needs to be changed here.

- The methods used in the experiment are described in great details and it is easy to find references and navigate. However, language needs improvement.

Validity of the findings

- Data points have been provided clearly, but needs more description like mentioned in point no. 1. However, statistically they are sound and controlled.

- Conclusions are very well stated and addressed with supporting results in form of figures and maps.

Additional comments

No comments

---

## Round 0.2 · Minor Revisions

Please make sure to cite the references correctly.

Reviewer 1 ·

Basic reporting

It is clearer now with sufficient background provided.

Experimental design

Methods described with sufficient detail.

Validity of the findings

Conclusions are well stated.

Additional comments

The authors have addressed most of my comments. The only minor point is Line 127 where the authors made wrong citations. The right ones were listed below.

Ma YP, Zhao L, Zhang WJ, Zhang YH, Xing X, Duan XX, Hu J, Harris AJ, Liu PL, Dai SL, Wen J. Origins of cultivars of Chrysanthemum—Evidence from the chloroplast genome and nuclear LFY gene. Journal of Systematics and Evolution, 2020, 58:925-944.

Qi S, Twyford AD, Ding JY, Borrell JS, Wang LZ, Ma YP, Wang N. Natural interploidy hybridization among the key taxa involved in the origin of horticultural chrysanthemums. Journal of Systematics and Evolution, 2021, doi: 10.1111/jse.12810.

---

## Round 0.3 · accepted · Accept

I have assessed the revision and authors have incorporated the suggestions.